# The instability mechanisms and precursor information of different type rocks based on acoustic emission

Ruixiong Xue[1,2]*, Yinghui Kong[1,2], Jinxiang Feng[1,2]

**1** School of Resources and Environmental Engineering, Inner Mongolia University of Technology, Hohhot, China, **2** Inner Mongolia Engineering Research Center of Geological Technology and Geotechnical Engineering, Inner Mongolia University of Technology, Hohhot, China

* xuerx@imut.edu.cn

## Abstract

This study presented the results of indoor uniaxial compression and acoustic emission (AE) monitoring tests for five type rocks: marble, granite, siltstone, sandstone and limestone. The experiment adopted a displacement-controlled loading method with a loading rate of 0.18mm/min, and AE monitoring was carried out synchronously. Some AE parameters were obtained. Detailed analysis of AE parameters RA and AF values revealed the evolutionary laws of the RA/AF value distribution range and rock failure mechanisms for five different type rocks; The models of damage types inside rocks over time had been established. Furthermore, the microseismic parameter lg$N$/$b$ was introduced. Analyzing the lg$N$/$b$ values dynamic change process revealed the instability laws, and rock instability precursor information was obtained. The results showed that an increase in the proportion of shear cracks, along with a wider range of the RA/AF values distribution can serve as precursor information for rock instability. Before rock failure occurred, the lg$N$/$b$ values consistently increased. Additionally, the study found that the change laws of lg$N$/$b$ values can reveal the extent of development of intrinsic weak structures inside the rocks. Through a multi-parameter and multi-method comprehensive analysis, we can more scientifically identify the instability precursor information of different type rocks in terms of time, and the hierarchical warning was achieved. The research results can provide a reference for microseismic monitoring at various sites and establish certain scientific basis for effective on-site warning strategies.

## 1. Introduction

With the development of the economy, the scale of infrastructure construction continues to expand, and the demand for energy is also constantly increasing. Numerous large-scale hydraulic and hydroelectric projects have begun to be constructed, investment in basic transportation construction has increased, and mineral resources

**Data availability statement:** All relevant data are within the manuscript and its Supporting Information files.

**Funding:** This work was supported by the National Natural Science Foundation of China (No. 42167023), and the Basic Research Funds for Universities Directly under the Inner Mongolia Autonomous Region of China (No. JY20230056) for which the authors are very grateful.

**Competing interests:** The authors have declared that no competing interests exist.

are gradually being developed deeper. The characteristics including long, large, deep and clustered, of underground engineering such as hydraulic tunnels, transportation tunnels, and mining tunnels are becoming increasingly prominent, leading to many rock mechanic problems such as large deformation, collapse, and rockburst of surrounding rock [1–3]. These problems seriously restrict the improvement of underground engineering construction level and pose a considerable threat to on-site workers and engineering equipment.

Rock is the basic material that makes up the earth's crust. It is a natural geological body formed by the aggregation of minerals or rock fragments according to certain laws under geological processes. In underground rock engineering, the surrounding rock mass that undergoes a change in stress state due to excavation is called surrounding rock. Rock instability describes the process in which the rock mass loses its established mechanical equilibrium, leading to ongoing deformation that progresses until failure occurs [4–6]. Understanding this process and its underlying causes is essential for accurately predicting and preventing rock instability. Numerous researchers have undertaken extensive studies from various perspectives [7–12]. Uniaxial and triaxial compression tests are commonly used to assess the compressive strength, deformation properties, and failure modes of materials, as well as engineering design and safety assessment. The integration of the testing approach with AE technology that can monitor changes within materials has been extensively adopted by researchers to monitor rock instability [13–15]. Zhang et al. [16] investigated failure characteristics and energy evolution process of delayed and instantaneous rockburst of basalt rock based on single-sided unloading experiments under true triaxial conditions by using high-speed photography and AE monitoring. Zhan et al. [17] conducted uniaxial compression tests on anchoring joint samples with different anchoring defects and obtained their rupture precursor characteristics. Zhang et al. [18] used infrared radiation (IR) and AE technologies to monitor the failure process of red sandstone during uniaxial loading experiments in real time, to quantify rock damage and predict failure from an energy perspective. Miao et al. [19] used the AE technique to examine variations in grain-size cracking behavior between specimens with tensile- and shear-induced fractures. There are notable differences in the AE characteristics between the shearing process and the stretching process. Yang et al. [20] conducted an experimental study on uniaxial compression of granite containing both planar and circular weak filling of different diameters with the aid of AE monitoring system, analyzed the mechanical behaviour and spatial fracture characteristics of weakly filled granite with different diameters under uniaxial loading. The combination of indoor testing and AE technology has become a well-established research method for analyzing the process of rock instability and extracting precursor information for early warning [21–25]. Liu et al. [26] explored the variation in rock AE characteristics with strain rate. A transition of a sudden increase in the RA-value and decrease in the AF-value occurs when the stress reaches a certain level. Sun et al. [27] studied AE *b*-value characteristics of rocks under different stress paths and the nonlinear characteristics and failure precursor laws of rocks were explored.

The instability laws and monitoring parameter thresholds of surrounding rock vary at different construction sites. In the early stage of construction, in order to better serve on-site construction, it is necessary to understand the instability process and mechanism of different type surrounding rocks in advance, and then formulate monitoring and warning plans. The paper aims to explore the same laws and different characteristics of instability processes and precursor information for different type rocks. Five different type rocks from various sources were selected for research and analysis. Based on AE technology, uniaxial compression tests were conducted on rocks under the same conditions to analyze the dynamic evolution process of the RA/AF values and lg$N$/$b$ values from the loading onset until instability occurs, thereby elucidating the instability mechanisms of different type rocks. Additionally, the study extracted precursor information of rock instability; the commonalities and differences of instability mechanisms and precursor information for different type rocks were explored. Multi-parameters and hierarchical early warning for different type rocks were achieved. We can take timely measures to save costs and improve construction efficiency. The research results have a certain reference value for monitoring and early warning of rock instability for different on-site construction, as well as for implementing effective scientific protective measures.

## 2. Test design

### 2.1. Test equipment

The test equipment primarily consists of a loading system and an AE monitoring system. For the test, the YAS-600 microcomputer-controlled electro-hydraulic servo uniaxial testing machine with high stiffness was chose as the loading system. This equipment is composed of a high-stiffness integral frame host, a servo-hydraulic power source, a digital control system, an oil cooling system, and a computer. It is capable of displaying the real-time status of the entire testing process and measuring the uniaxial compressive strength of standard rock samples, while also drawing the stress-strain curves of rock samples. The AE system employs the PCI-2 multi-channel monitoring system, manufactured by Physical Acoustics Corporation (PAC) in the United States. The trigger threshold was set to 40 dB, and the sampling rate was set to 5 MSPS. The preamplifier gain is 40dB, and the filtering range is 20–100 kHz. The values of PDT, HDT and HLT were set at 50 us, 200 us and 300 us respectively. The experiment adopted a displacement-controlled loading method with a loading rate of 0.18mm/min, and AE monitoring was carried out synchronously.

### 2.2. Specimen preparation

This study includes five different type rock samples: granite, marble, sandstone, siltstone, and limestone, separately sourced from locations including Suizhou in Hubei, Fangshan in Beijing, Zigong in Sichuan, Ordos in Inner Mongolia, and Jingxing in Hebei. They were labeled as HG, DL, S, FS, and SH, as shown in Fig 1. The samples are standard cylindrical specimens, with a diameter of 50 mm and a height of 100 mm. The machining accuracy complies with the relevant standards. The non-parallelism of both end faces of the rock samples is maintained below 0.2 mm to avoid local stress concentration caused by bias pressure affecting the test results. Three samples were tested for each type rock, and due to limitations in the length of the article, a typical sample was selected for analysis for each type rock samples. The sample with compressive strength closest to the mean of similar samples was as the typical sample.

## 3. Analysis of AE parameters for different type rocks

### 3.1. Evolutionary characteristics of the RA/AF values and crack proportions

Numerous studies [28,29] have shown that the RA (rise time/amplitude) and AF (AE counts/duration) ratio can effectively characterize the types of rock damage and reveal the mechanisms of rock instability. A low AF value and a high RA value indicate the generation and development of shear cracks, whereas the opposite conditions suggest the generation and development of tensile cracks. This study employed the RA/AF ratio ($k$) to differentiate between tensile and shear cracks. When the value of $k$ exceeds 0.05, it signifies the formation of shear cracks inside the rock; otherwise, it indicates the generation of tensile cracks [29].

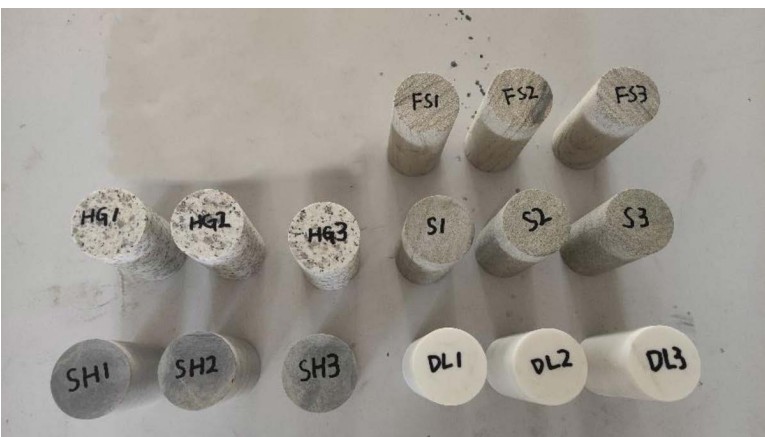

**Fig 1. Rock samples.**

Fig 2 shows the RA/AF values distribution and the proportion of crack types of AE signals during the loading process of marble. In Fig 2B, the horizontal axis represents the time period numbers, with each period corresponding to a duration of 20 seconds. The blue line in the figure marks the time point when the distribution range of RA/AF values significantly widens, while the green line denotes the time point of rock instability. In the paper, the blue arrow represents the first warning time, the yellow arrow represents the second warning time, and the red arrow represents the rock failure time.

Fig 2 shows that the marble experienced a total duration of 300 seconds from the initial loading to the point of instability, divided into 15 distinct time segments. During time periods 1–11, the proportions of shear and tensile cracks remain relatively stable with shear cracks accounting for 22% to 33%, and the distribution of RA/AF values in the range of 2 ~ 5ms/V·kHz was relatively small, which indicated that the rock was in the stage of elastic deformation to stable development of micro elastic cracks, indicating rock stability was relatively good. As loading continues into time period 12 (from 220s to 240s), the proportion of shear cracks began to increase, reaching 38%, accompanied by a significant increase in data points within the RA/AF value range of 2–5ms/V·kHz. As the loading continued, the proportion of shear cracks continued to increase, reaching a peak of 56%. As the loading progressed, the proportion of shear cracks began to decrease, but the value remained relatively high. Ultimately, when the loading reaches 300s, the main cracks inside the rock formed, resulting in rock instability. Based on the analysis above, we can consider time periods 12–14 as a warning period which was a stage of unstable fracture development for the rock. The endpoints of time periods 12 and 14 could be used as the first and second warning points, respectively. Notably, the second warning point was located approximately 20 seconds before the instability point, representing 6.7% of the entire loading process. Hierarchical warning during the rock loading process can achieve a good warning effect. It can also be observed that the transition of rock from a stable to an unstable state corresponds to a transformation in the damage types occurring inside the rock. The findings of the research can provide a guidance for on-site construction, ensuring that it is carried out safely and efficiently.

Fig 3 shows that the granite experienced a total duration of 422 seconds from the initial loading to the point of instability, divided into 22 distinct time segments. The figure also revealed that the instability mechanism of the granite sample during from the initial loading to instability was quite complex, with the proportions of damage crack types being unstable and exhibiting relatively large fluctuations. From the initial loading stage to the warning stage, the proportion of tensile cracks changed in an M-shaped variation pattern. During the time periods 3–8, the proportion of shear cracks gradually increased at a relatively slow rate for a longer period, ultimately reaching 52% which was relatively large; the main distribution range of RA/AF values was relatively narrow and RA/AF values were relatively small. As the loading continued, the proportion of shear cracks gradually decreases, reaching 20% which was relatively small. Subsequently, the proportion of

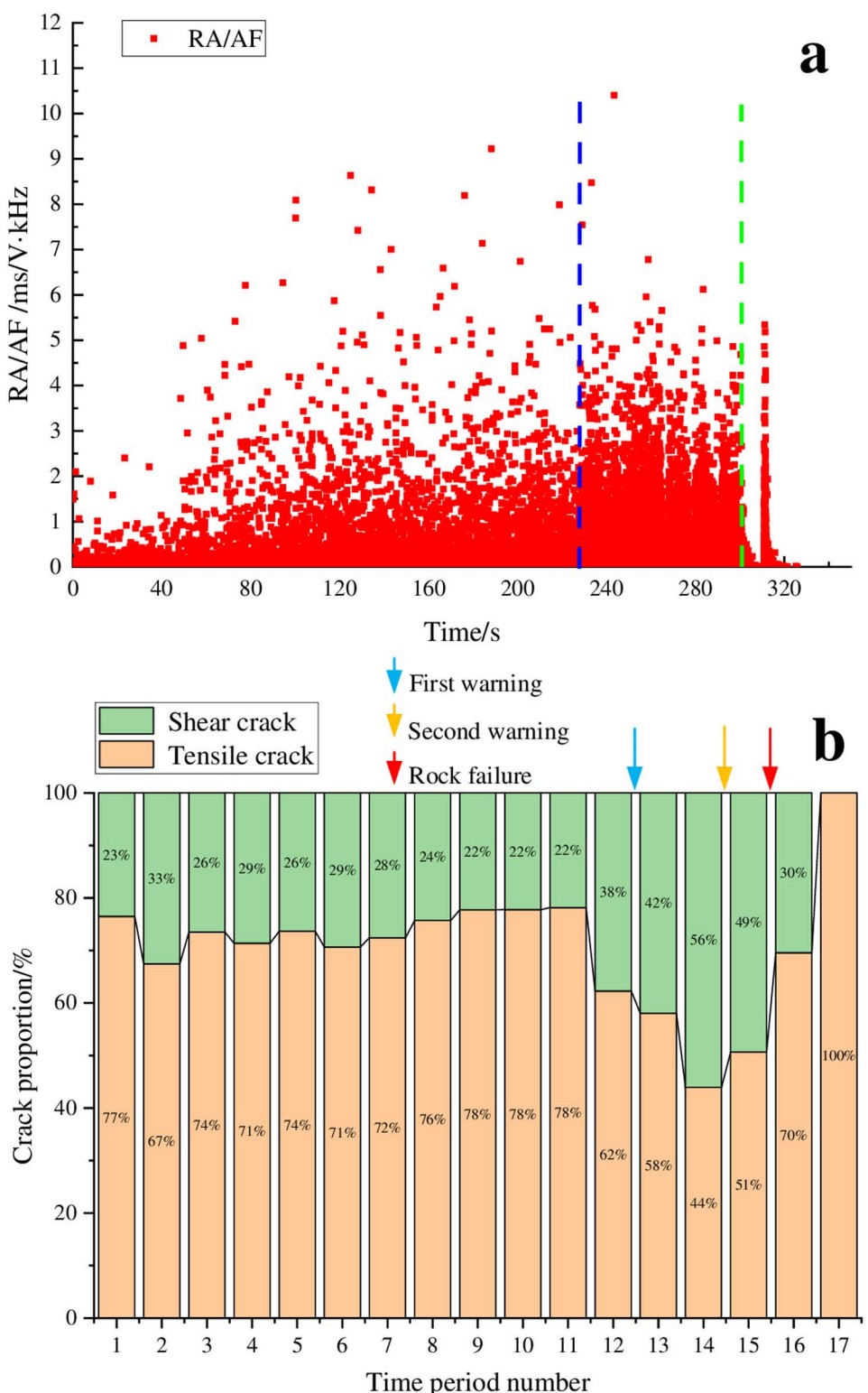

**Fig 2. RA/AF value distribution and proportion of cracks for the marble sample.** (a) RA/AF value distribution, (b) Proportion of tensile and shear cracks.

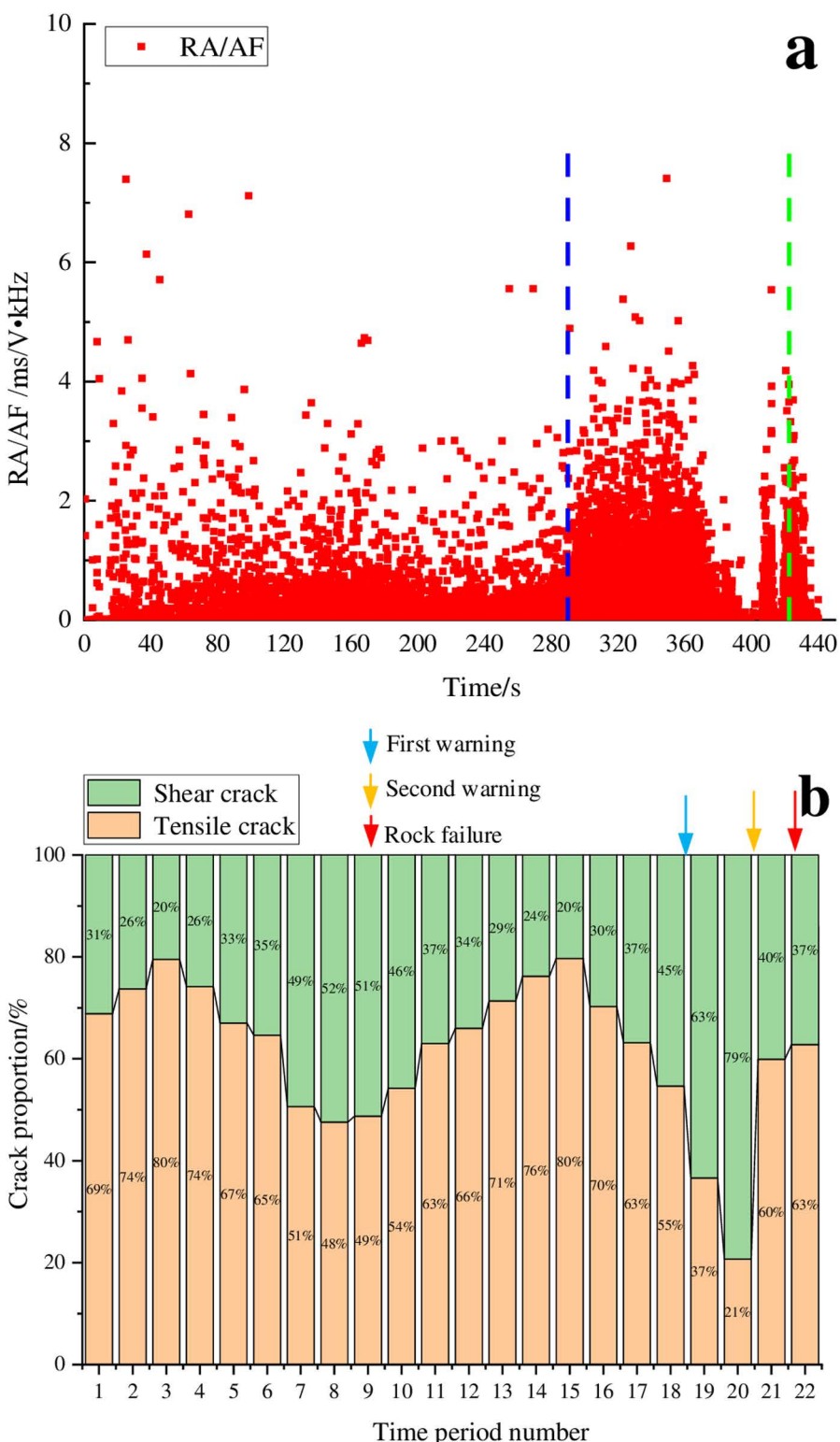

**Fig 3. RA/AF value distribution and proportion of cracks for the granite sample.** (a) RA/AF value distribution, (b) Proportion of tensile and shear cracks.

shear cracks increased again, experiencing a sharp increase during time periods 18–20. The proportion of shear cracks reached a maximum of 79%, and then rapidly decreased until the rock became unstable. Based on the observed changes in the proportion of shear cracks, the failure process of granite was relatively complex, leading to early warning more challenging. We can consider time periods 18–20 as the stage of unstable rock fracture development, and the end point of time period 20 as the warning point which was approximately 22 seconds (accounting for 5.2% of the entire loading process) prior to the instability point. Granite instability manifested as a bursting phenomenon, characterized by relatively loud noises and the ejection of segments. Consequently, on-site technicians can evaluate the stability of the surrounding rock by monitoring and sounds.

Fig 4 shows that the siltstone experienced a total duration of 436 seconds from the initial loading to the point of instability, divided into 22 distinct time segments. The figure shows that the entire loading process of the siltstone can be divided into three distinct stages. During time periods 1–11, there was significant fluctuation in the proportions of shear cracks and tensile cracks. The proportion of shear cracks first gradually increased, and then gradually decreased, with the maximum proportion of shear cracks reaching 41%; the RA/AF values were mainly distributed in the range of 0 to 0.5 ms/V·kHz. As the loading progressed, during the time periods 12–18, the proportions of shear cracks remained relatively stable at 11% to 15%, and the RA/AF values were also relatively small, less than 2 ms/V·kHz. With further loading, the proportion of shear cracks began to increase, and the RA/AF values also significantly increased with obvious clustering in the region which was greater than 3 ms/V·kHz. After a thorough analysis, we can consider the endpoints of time periods 19 and 21 as the first and second warning points, respectively. Notably, the second warning point was located approximately 16 seconds before the instability point, representing 3.7% of the entire loading process. Hierarchical warning during rock loading can achieve a good warning effect.

Fig 5 shows that the sandstone experienced a total duration of 408 seconds from the initial loading to the point of instability, divided into 21 distinct time segments. This loading process can be separated into two distinct stages. The first stage included time periods 1–17, during which the proportion of shear cracks initially increased and then decreased, ranging from 10% to 33%. The RA/AF values were mainly concentrated in the range of 0–2 ms/V·kHz, with a denser cluster in the range of 0 to 0.5 ms/V·kHz. As the loading progressed, the proportion of shear cracks increased once more at a faster rate, and the distribution range of RA/AF values widened with more points distributed in the range of 1–5 ms/V·kHz. After a comprehensive analysis, the endpoint of the period 18 can be identified as the first warning point. Subsequently, the endpoint of the period 20 was identified as the second warning point, indicating that rock instability was anticipated in about 8 seconds which accounted for 2% of the entire loading process.

Fig 6 shows that the limestone experienced a total duration of 272 seconds from the initial loading to the point of instability, divided into 14 distinct time segments. This loading process can be separated into two distinct stages. During time periods 1–8, the proportion of shear cracks gradually increased to 45% and then gradually decreased. For most of the time, the proportion of shear cracks remained below 40%, and the RA/AF values were mainly concentrated in the range of 0–1 ms/V·kHz. As loading progressed into the period 9, the proportion of shear cracks experienced a rapid increase, reaching 49%. From time period 11, the proportion of shear cracks had remained above 40%, the main distribution range of RA/AF values had significantly widened, and the more data points were found between 1–6 ms/V·kHz, along with some points appearing in the range of 6–10 ms/V·kHz. Through a comprehensive analysis, we identified the 220 s as the first warning point and the 260 s as the second warning point. The second warning point was located approximately 12 seconds before the instability point, representing 4.4% of the entire loading process.

### 3.2. Analysis of the lg*N*/*b* values

The *b* value was initially used to describe the proportional relationship between the earthquake magnitude and frequency, with its calculation expressed by the well-known Gutenberg-Richter (G-R) relationship [30]:

$$\lg n = a - bM \tag{1}$$

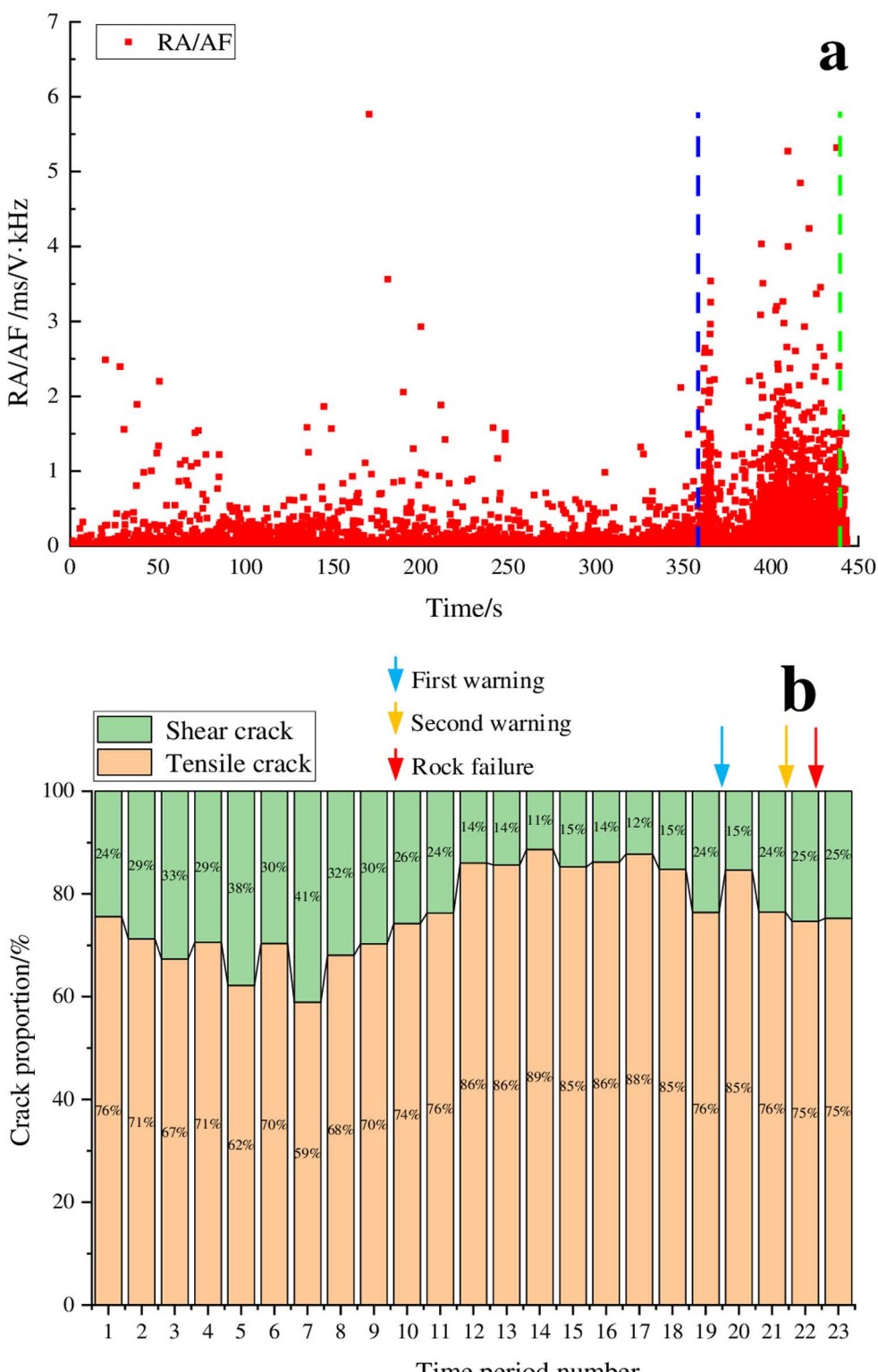

**Fig 4. RA/AF value distribution and proportion of cracks for the siltstone sample.** (a) RA/AF value distribution, (b) Proportion of tensile and shear cracks.

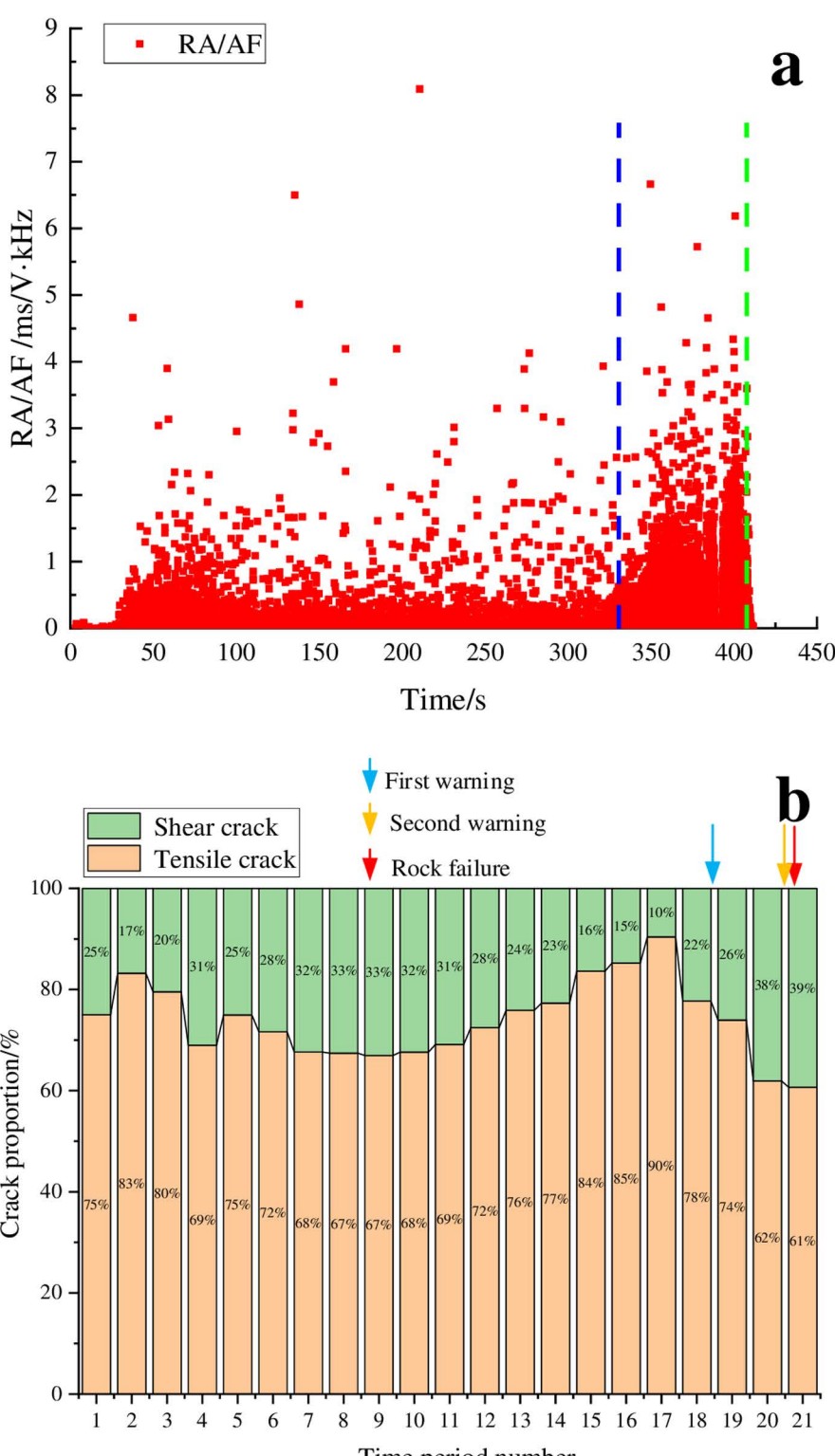

**Fig 5. RA/AF value distribution and proportion of cracks for the sandstone sample.** (a) RA/AF value distribution, (a) Proportion of tensile and shear cracks.

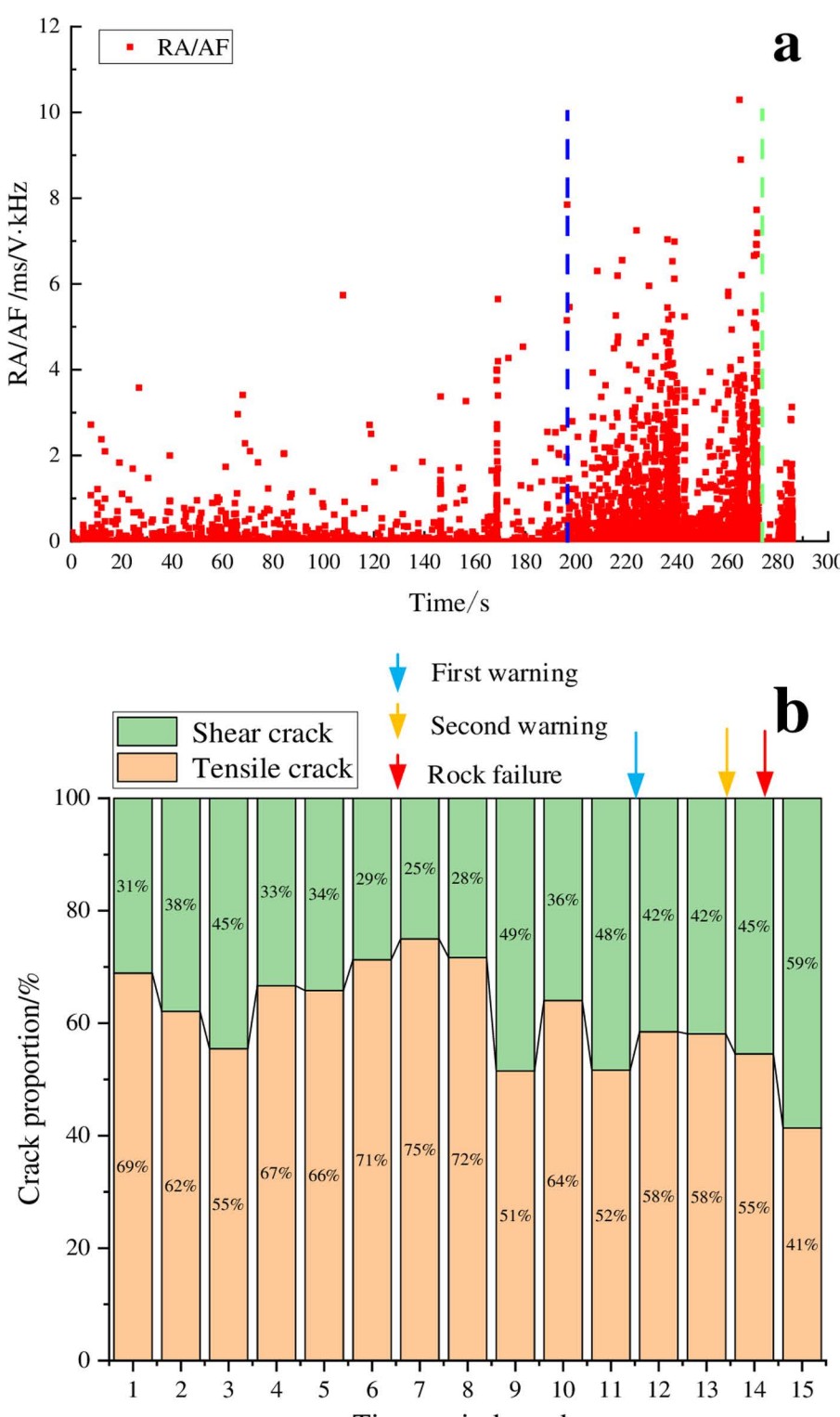

**Fig 6. RA/AF value distribution and proportion of cracks for the limestone sample.** (a) RA/AF value distribution, (b) Proportion of tensile and shear cracks.

where $M$ is the magnitude, $n$ is the number of earthquakes whose magnitude is no less than $M$, and $a$ and $b$ are constants.

Based on the principle that AE elastic waves and seismic waves are similar to the accompanying phenomena of material damage, the AE events captured during rock failure process can be regarded as seismic activity, and the $b$-value can be studied to analyze the characteristics of rock deformation and failure. By adjusting the G-R relationship, we substitute the magnitude with the AE amplitude. The modified formula is presented as equation (2) [27].

$$\lg n = a - b\left(\frac{A_{dB}}{20}\right)$$ (2)

where $A_{dB}$ is the amplitude of the AE signal; $n$ is the number of AE hits whose amplitude is no less than $A_{dB}/20$; $b$ is the slope of the line obtained by linearly fitting the curve using the least squares method.

The $b$ value represents the proportional relationship between the number of large magnitude events and the number of small magnitude events. Therefore, the change in the $b$ value can be used to reflect the change in the stress field of the surrounding rock mass. The larger the number of events $N$, the more microcracks will appear in the surrounding rock mass, and the greater the risk of surrounding rock mass instability. Xue et al. [31] introduced $\lg N/b$ as a novel indicator for analyzing precursor information of rock instability, demonstrating effective predictive performance. Therefore, this paper introduced the microseismic parameter $\lg N/b$ to evaluate the stability and instability risk of rocks during the loading process. As illustrated in the figure below, the analysis of the rock loading process was conducted by plotting the change curve of $\lg N/b$ with a 20 second sliding window.

In Fig 7, during the early loading stage, for the marble, the $\lg N/b$ values first decreased, then fluctuated around 1.8, and subsequently decreased again; the overall fluctuation of the curve was relatively small. This indicated that the primary cracks inside the rock were rapidly compacted under stress, and a significant number of low-energy hits occurred. Beginning from time period 6, the $\lg N/b$ values increased quickly indicating that a large number of new microcracks began to form inside the rock along with slippage and displacement of crystal particles. During the time period 14, the $\lg N/b$ values reached the peak, with a peak value of 2.73; at this time, the AE activity inside the rock was most active, the internal cracks of the rock were interconnected forming large-scale macroscopic fractures, and the rock bearing capacity was close to the threshold. Subsequently, the $\lg N/b$ values decreased and instability occurred in time period 15. Based on the dynamic evolution characteristics of $\lg N/b$ values and their specific physical meanings, the peak timing of the $\lg N/b$ values can be used as warning points. At this time, the AE activity inside the rock became more intense, with a large number of AE hits including more high-energy hits. Although failure surfaces had formed, the rock still possessed some load-bearing capacity, with about 20 seconds left before instability and failure, indicating a significant predictive capability.

At the loading onset, the granite contained only a few microcracks. As the loading increased, these microcracks progressively closed. In Fig 8, the $\lg N/b$ values fluctuated between 1 and 2 with relatively small fluctuations, indicating weak microseismic activity inside the rock and good rock stability. From time period 13, the $\lg N/b$ values began to increased gradually indicating that some new cracks gradually generated and expanded. Starting from the time period 16, the $\lg N/b$ values exhibited a significant increase, peaking at 7.19 during time period 20 during which the cracks began to interconnect. Subsequently, the $\lg N/b$ values experienced a rapid decrease, and during time period 22, the rock experienced instability and failure. The time period 20, which was 22 seconds away from rock instability, was designated as a warning point. During the time period 20, the curve reached its peak, indicating a sharp increase in internal rock damage and the formation of numerous relatively large fractures. At this time, an early warning can optimize both the safety and efficiency of construction activities.

In Fig 9, during the initial stage of loading, the $\lg N/b$ values first gradually decreased and then gradually increased, maintaining a continuous sharp rise and fall phenomenon for a period of time. Previous studies [32] have shown that siltstone is a porous and weakly cemented rock. During the initial application of load, the closure of the rock's pores, and

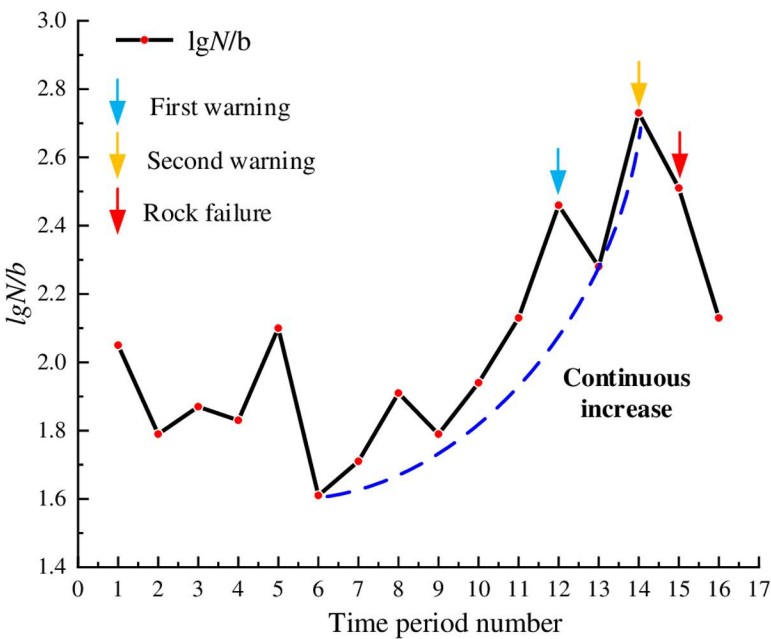

**Fig 7. Temporal distribution of lg*N*/*b* values for marble.**

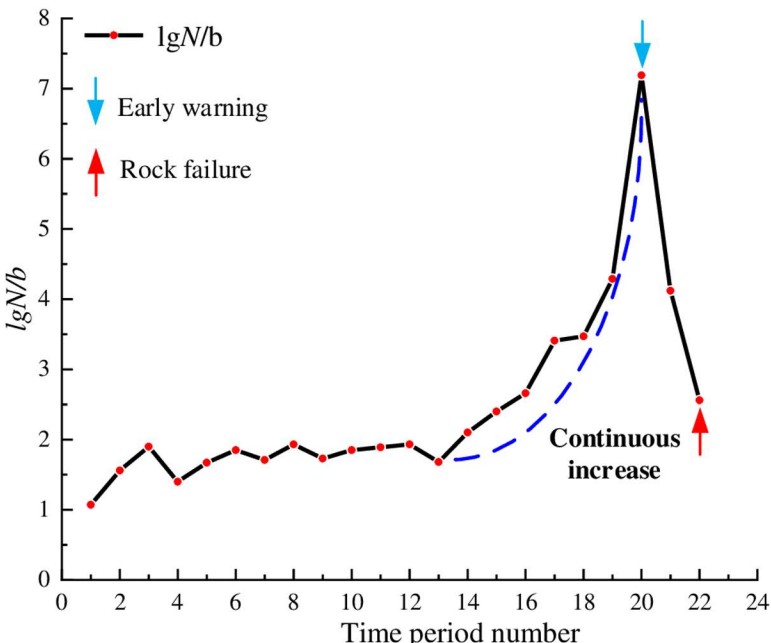

**Fig 8. Temporal distribution of lg*N*/*b* values for granite.**

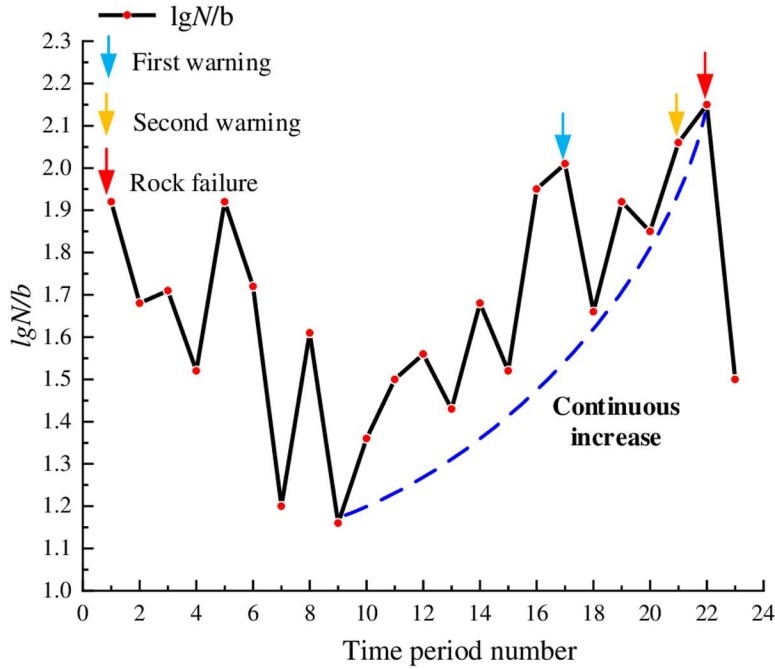

**Fig 9. Temporal distribution of lg*N*/b values for siltstone.**

the uneven scales and patterns of weak structural failure resulted in significant fluctuations and a downward trend of the lg*N*/b values. Subsequently, although the lg*N*/b values continued to fluctuate up and down, the overall trend was gradually increasing. At this time, the primary cracks were compacted, new cracks began to generate and develop, and the internal damage of the rock began to accumulate and increase. During the 16 and 17 time periods, the lg*N*/b values continued to increase, indicating that the AE activity was quite active and there were relatively large cracks forming inside the rock. Subsequently, the curve experienced a sharp decline. From time period 19, the curve rose again, which indicated the risk of rock instability increased; this rise reached another peak during time period 22; during the curve ascent process, the rock bearing capacity gradually approached the threshold. Through a comprehensive analysis, we can designate the end-point of the time period 17 as the first warning point and the endpoint of the time period 21 as the second warning point. During the time period 21, the lg*N*/b value was the relatively large before the failure, and there were many high-energy AE hits inside the rock.

In Fig 10, in the early loading stage of sandstone, the lg*N*/b values were relatively low, first increased and then gradually decreased, which was the process of microcrack generating. During the time periods 17 and 18, the lg*N*/b values sharply increased, reaching a peak of 3.28; this indicated that larger-scale fractures inside the rock sharply increased and the AE signal had entered an active stage. Following a minor decline, the curve peaked once more during the time period 20, during which macro-fractures quickly emerged on the rock surface and microcracks started to interconnect to create a failure surface. Subsequently, the curve declined and the rock became unstable 8 seconds later. Therefore, the endpoints of the time periods 18 and 20 were appropriate to serve as the first and second warning points, respectively. Both time markers corresponded to peak points on the curve, indicating that there were more relatively high-energy AE hits inside the rock.

Limestone belonged to rocks with horizontal bedding structures and fewer primary fractures. In Fig 11, the lg*N*/b values fluctuated slightly during the initial loading stage, predominantly ranging from 0.5 to 1.5. After the internal microcracks of

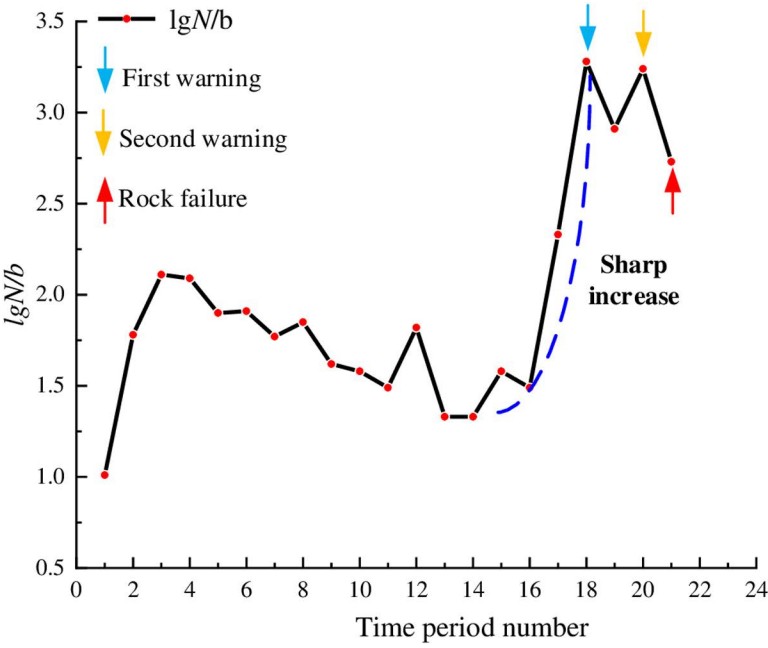

**Fig 10. Temporal distribution of lg$N$/b values for sandstone.**

the rock were compacted, the lg$N$/$b$ values gradually increased from the 8th time period. This trend indicated the stable development of microcracks and the significant accumulation of elastic potential energy within the rock. Starting from time period 11, the lg$N$/$b$ values showed a substantial increase, peaking at 3.82 during the time period 13. At this time, the elastic potential energy accumulated within the rock was quickly released; numerous macrocracks appeared on the rock surface; the microcracks inside the rock developed into larger fractures, forming failure surfaces, and the rock's load-bearing capacity approached its threshold. Consequently, rock failure occurred during the time period 14, and the lg$N$/$b$ values rapidly declined. By identifying the curve peak points as warning points, it can be noted that rock instability occurred at approximately 12 seconds later. The warning effect was good.

By analyzing the lg$N$/$b$ values of the five type rocks mentioned above, it can be concluded that the lg$N$/$b$ values can effectively reflect the internal AE activity characteristics of rocks from the initial loading to instability, and can well characterize the process of micro-fracture generate, development, and expansion inside rocks. The warning effect was good.

## 4. Discussion

This paper conducted indoor AE tests on five different type rocks subjected to uniaxial loading, with a constant loading rate of 0.18 mm/min. Due to the variations in mineral composition and grain size among these rock types, they displayed distinct AE characteristics and diverse mechanical responses during the loading process. Thus, the comprehensive analysis of the instability mechanisms and the extraction of precursors to instability for different type rocks are both a key area of interest and a significant challenge.

First, under uniform experimental conditions, different type rocks exhibited different durations from the initial loading to the instability: marble lasted 300 seconds, granite lasted 422 seconds, siltstone lasted 436 seconds, sandstone lasted 408 seconds, and limestone lasted 272 seconds. This also indicated that different type rock had different abilities to resist external disturbances. The distribution ranges and evolutionary laws of the RA/AF values were different. During the initial loading stage, Granite and marble were monitored to have a large number of AE hits, and the RA/AF values were

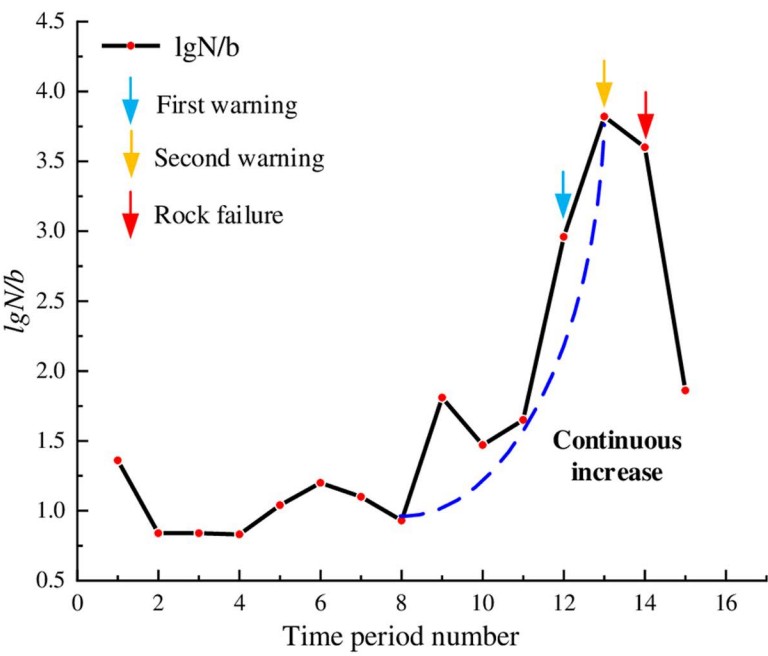

**Fig 11. Temporal distribution of lg*N/b* values for limestone.**

relatively high. From the proportion model of shear cracks and tensile cracks, it can be seen that the failure mechanism of the granite was the most complex, and the proportion of shear cracks significantly fluctuated during the initial loading stage. The evolution laws of lg*N/b* values were different. The lg*N/b* value curves of siltstone and sandstone fluctuate greatly; especially during the initial loading stage for siltstone, the lg*N/b* values continued to decrease for a long time. Through research, it had been found that siltstone is a porous and weakly cemented rock, with relatively developed primary fractures. Therefore, the evolution laws of lg*N/b* values can indirectly reflect the structural characteristics inside the primary rocks.

A comprehensive analysis revealed common precursor characteristics among all five type rocks, including an increase in the proportion of shear cracks and a wider distribution range of the RA/AF values, and continuously increasing in the lg*N/b* values [29,31]. Due to the relatively complex instability mechanism and process of rocks, we need to analyze their multiple parameters and hierarchical warning was achieved as much as possible, in order to take timely measures and achieve optimal construction results.

Managing timing and establishing thresholds present significant challenges in instability warnings at this stage; nonetheless, they remain focal points, especially critical for on-site construction. For example, during the construction process, the key issues that researchers and on-site technicians need to address are what measures to take and when to evacuate the site when encountering danger. A thorough analysis of the time to the instability of the rock and the rock failure process is conducted to enable a tiered warning system for rock instability and establish warning levels. In this study, the five type rocks, namely marble, granite, siltstone, sandstone and limestone, showed that the time from the final warning point to the instability point accounted for 6.7%, 5.2%, 3.7%, 2%, and 4.4% of the total loading time, respectively.

Different type rocks exhibited different durations from the initial loading to the instability. On site, under the same working conditions, different types of rocks have different resistance to disturbance, and the measures taken are different. When different types of rocks fail, the parameter threshold is different, while the change trend is consistent, which can be used as a reference for on-site monitoring. On site, it is difficult to complete different type rock failure monitoring tests.

At the same time, we have developed hierarchical warning time points for different type rocks based on AE parameters. When designing safety protocols, risks can be evaluated and emergency plans can be developed based on different warning points to ensure the safe evacuation of on-site personnel and equipment.

This study set the time periods for AE parameters at 20 seconds. To enhance time resolution, a shorter time window can be utilized to fulfill warning requirements; however, this will lead to an increase in computation time.

## 5. Conclusions

In the paper, uniaxial compression tests were conducted on five different type rocks. The evolution characteristics of AE parameters RA/AF and lg$N$/$b$ values were analyzed in detail, and precursor information of instability for different rocks were extracted. The research results can achieve multi-parameter and hierarchical early warning. The main conclusions are as follows:

(1) The proportion of shear cracks inside rocks increased and the distribution range of RA/AF values correspondingly widened, which can be used as precursor information for instability for all type rocks. There are notable differences in the RA/AF values and damage evolutionary characteristics for different type rocks. During the initial loading stage, Granite and marble were monitored to have a large number of AE hits, and the RA/AF values were relatively high. Additionally, the failure mechanisms of granite and siltstone were more complex compared to those of the other type rocks.

(2) The continuous and rapid increase in lg$N$/$b$ values can serve as instability precursor information for all type rocks. The evolution laws of lg$N$/$b$ values varied for different type rocks. The lg$N$/$b$ value curves of siltstone and sandstone fluctuate greatly; especially during the initial loading stage for siltstone, the lg$N$/$b$ values continued to decrease for a long time. Through research, it had been found that siltstone is a porous and weakly cemented rock, with relatively developed primary fractures. Before the instability and failure of the rock, the lg$N$/$b$ values consistently increased. The variations in AE characteristics aligned well with the evolution process of the sample damage. Different rock types exhibited varying lg$N$/$b$ threshold values. Additionally, the change laws of lg$N$/$b$ values can also reveal the development degree of primary weak structures inside rocks.

(3) The warning methods in the paper can achieve the goal of multi-parameter and hierarchical warning for rock instability. A thorough analysis of the instability timing and the failure processes of the rock, we have summarized some laws. During the later loading stage, the proportion of shear cracks sharply increased and the distribution range of RA/AF values was relatively wide, which can be used as a warning point; if the proportion of shear cracks continued to remain at a relatively high level, it can be used as the second warning point. Furthermore, the time nodes at which the lg$N$/$b$ values continuously and rapidly increased and reached the peak point for the first time and again were considered as the first and second warning points. In this study, the five type rocks, namely marble, granite, siltstone, sandstone and limestone, showed that the time from the final warning point to the instability point accounted for 6.7%, 5.2%, 3.7%, 2%, and 4.4% of the total loading time, respectively.

The research results provide a scientific basis for scientific early warning and prevention of on-site rock instability. The on-site environment is very complex, and sensors can receive a lot of noise and the signal also contains some noise; The laboratory environment is relatively simple. The rock damage also includes unloading damage on site. Blasting shock and structural planes can also affect the failure mechanism of rocks. All of the above are the direction and focus of our future research. Based on the on-site complexity, the combination of AE monitoring and intelligent warning has great advantages, which will also be the focus of our future research.

## Supporting information

**S1 Dataset. Minimal data set.**
(ZIP)

## Author contributions

**Conceptualization:** Yinghui Kong.

**Data curation:** Yinghui Kong, Jinxiang Feng.

**Funding acquisition:** Ruixiong Xue.

**Investigation:** Ruixiong Xue.

**Methodology:** Ruixiong Xue, Yinghui Kong.

**Project administration:** Ruixiong Xue.

**Resources:** Ruixiong Xue.

**Software:** Yinghui Kong, Jinxiang Feng.

**Writing – original draft:** Ruixiong Xue, Yinghui Kong.

**Writing – review & editing:** Ruixiong Xue, Yinghui Kong.

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
