## [Decision Letter · Decision Letter 0]

28 Jan 2025

PONE-D-25-00922The instability mechanisms and precursor information of different type rocks based on acoustic emissionPLOS ONE

Dear Dr. Xue,

Thank you for submitting your manuscript to PLOS ONE. After careful consideration, we feel that it has merit but does not fully meet PLOS ONE’s publication criteria as it currently stands. Therefore, we invite you to submit a revised version of the manuscript that addresses the points raised during the review process.

We look forward to receiving your revised manuscript.

Kind regards,

Kang Wang, Ph.D.

Academic Editor

PLOS ONE

**Journal Requirements:**

Reviewers' comments:

Reviewer's Responses to Questions

**Comments to the Author**

1. Is the manuscript technically sound, and do the data support the conclusions?

Reviewer #1: Yes

Reviewer #2: Yes

Reviewer #3: Yes

Reviewer #4: Partly

Reviewer #5: Yes

Reviewer #6: Yes

2. Has the statistical analysis been performed appropriately and rigorously? 

Reviewer #1: Yes

Reviewer #2: Yes

Reviewer #3: Yes

Reviewer #4: Yes

Reviewer #5: No

Reviewer #6: Yes

3. Have the authors made all data underlying the findings in their manuscript fully available?

Reviewer #1: Yes

Reviewer #2: Yes

Reviewer #3: Yes

Reviewer #4: Yes

Reviewer #5: No

Reviewer #6: Yes

4. Is the manuscript presented in an intelligible fashion and written in standard English?

Reviewer #1: Yes

Reviewer #2: No

Reviewer #3: Yes

Reviewer #4: Yes

Reviewer #5: No

Reviewer #6: Yes

5. Review Comments to the Author

**Reviewer #1:**  This study investigated the instability mechanisms and precursor information of five rock types—marble, granite, siltstone, sandstone, and limestone—based on uniaxial compression tests combined with acoustic emission (AE) monitoring. However, enhancing the clarity of the methodology, broadening the discussion's practical context, and providing further justification for certain methodological choices would strengthen the paper.

1、 Condense the abstract by focusing on key findings and their significance. The current version is dense and includes excessive technical details, which could be streamlined to enhance readability.

2、 Highlight the novelty of your study more explicitly. What sets your research apart from previous studies on rock instability and acoustic emission monitoring?

3、 Provide more detail on the rationale behind using RA/AF and lgN/b parameters for instability prediction. Briefly explain why these metrics are superior or complementary to other methods.

4、 While the introduction explains the significance of the study, it could benefit from more in-depth references to recent research in the past five years. Highlight gaps in the literature this study addresses, such as: compgeo.2024.106095; tafmec.2024.104691; fuel.2023.129584

5、 Clarify the selection criteria for the five rock types. Was it based on geographic diversity, mechanical properties, or application relevance?

6、 The results are well-documented but could benefit from clearer visual aids. Simplify or reorganize some graphs for better comprehension.

7、 Compare your findings with similar studies in more detail, emphasizing agreement or divergence. This contextualization will strengthen the discussion.

8、 Reiterate the practical significance of your findings, such as how they might influence on-site monitoring or the design of safety protocols.

9、 Discuss the limitations of your study and potential future work, such as testing under field conditions or extending the methodology to other rock types.

**Reviewer #2: ** This manuscript studies the instability mechanism and precursor information of different types of rocks based on acoustic emission. This topic is very important, and the author has meaningful insights on it. The structure of the manuscript is reasonable, and the content is also rich, but the research depth is slightly insufficient. Minor revision is recommended.

1. There are some grammatical errors in this manuscript, and an improvement of the English editing is needed.

2. What is the sensor type selected for the AE system?

3. Please explain the meaning of RA and AF values when their first appearance in the manuscript.

4. Please explain the meaning of “N” and “n” in the manuscript. Do the two letters have the same meaning?

5. The author's understanding of the rock fracture mechanism is somewhat lacking. It is recommended to refer to the following literature for further analysis

Analytical solution of the stress field and plastic zone at the tip of a closed crack. FRONTIERS IN EARTH SCIENCE. Doi: 10.3389/feart.2024.1370672

**Reviewer #3: ** Authors have meaningful insights into the topic. This is a research work with practical engineering significance. It is my opinion that this manuscript can be accepted for publication after considering the following issues.

1�The abstract should accurately reflect the research purpose, research methods, research results, and conclusions of the paper. It is recommended to rewrite it.

2�Although the introduction provides a literature review, it is not easy to truly grasp the work's novelty for the experimental study on rock mechanics. The aims and novelties should be explicitly mentioned in the introduction. It is suggested that the author supplement the research results on RA/AF and b values of rocks.

3�The key words selected in the paper are not accurate enough and representative, for example, rock mechanics, uniaxial compression, acoustic emission characteristics, early warning.

4�In line 96,“The equipment is currently recognized as one of the most advanced and feature-rich uniaxial testing machines for rocks available in China”, I do not recommend evaluating experimental equipment.

5�The full text suggests the author to refine the language expression, such as the first and third paragraphs of the introduction, equipment description, etc.

6�The parameter Settings of the acoustic emission acquisition system, such as sensor resonant frequency, HLT, HDT, PDT, etc., should also be provided.

7�In Fig.4, Please explain that there is no second warning point for granite.

8�Why did Fig.8, Fig.9, and Fig12 fail to get the second warning point? Does it indicate that the sliding window selected by the author is too large, resulting in inaccurate calculation results of b value?

9�The first warning points of marble and siltstone are different under the two methods� what is the reason?

10�It is suggested that when analyzing the variation characteristics of RA/AF value and lg N/ b, the author finally summarizes, analyzes and compares the difference of the precursor law of failure of five different lithologies.

11�There are some grammatical errors in this manuscript, and an improvement is needed.

12�“lgN/b values can effectively reflect the internal AE activity characteristics of rocks”. What is the physical meaning of the parameter?

13� Section “Analysis of AE parameters for different type rocks”. The meaning of RA and AF values need to be explained.

14�How to achieve synchronization for time control of AE monitoring system and time control of the loading system?

15� Line 238, add a reference to explain the source of the equation (2).

**Reviewer #4: ** The study aims to identify the type of failure in rock structures through wave characteristic through analysis of acoustic emissions (AE) monitoring data. The real-life problem it seeks to address is understanding the deformations in rock structures during various engineering processes. However, this aspect is not clearly explained in the rationale section.

Furthermore, it seems that the study assumes the reader has a solid understanding of AE monitoring, RA and RF analyses, and instability mechanism analyses. This section could benefit from more detailed explanations.

However, most problematic issue lies in the selection of rock samples. The study worked samples from various origins, which raises concerns about the potential for non-correlatable textural characteristics. For instance, void ratio is an important factor influencing deformation behavior in each rock type studied. Yet, the study neglected to consider important textural properties like void ratio and grain size. Additionally, in metamorphic rocks, matrix characteristics will also cast doubt on the results of an AE monitoring-based study with a limited number of samples from different rock types.

To address these concerns, it would be more meaningful for the study to conduct a controlled experimental process on rocks from the same group (e.g., granites with varying grain sizes). This approach would ensure that the results are more reliable and representative.

**Reviewer #5: ** General comments

1. The paper presents a thorough analysis of AE parameters across different rock types, providing valuable insights into their mechanical behavior and instability precursors. However, the paper would benefit from a more structured organization. Specifically, the authors should consider delineating the content more clearly by including a separate "2. Methodology" section that details experimental setup, data collection, and analysis methods, and a "5. Results" section that distinctly presents the findings before discussing their implications. This separation will aid in readability and help readers distinguish between the processes used and the outcomes observed.

2. Language and Grammar: The manuscript requires a thorough review for language and grammar improvements. The current draft has instances of awkward phrasing and grammatical errors that could hinder comprehension and detract from the professional quality of the publication. Employing a native English speaker for proofreading or utilizing professional scientific editing services could significantly enhance the clarity and flow of the text.

3. While the paper is rich in technical content, it occasionally assumes a level of pre-knowledge that not all readers might possess. The authors are advised to define all technical terms and acronyms at their first occurrence and ensure that explanations of complex concepts are accessible to a broader scientific audience without sacrificing depth.

4. The use of figures is commendable for illustrating complex data; however, the authors should ensure that all graphical representations are clear, well-captioned, and directly referenced in the text. Each figure should be accompanied by a descriptive legend that explains all symbols, scales, and colors used, even if these details are also discussed in the text.

5. It is recommended that the authors leave no subtitles blank throughout the document. Each section and subsection should have a descriptive heading that accurately reflects the content that follows. This approach not only improves the manuscript’s navigation but also aligns with academic standards for formal scientific reporting.

6. The discussion section would benefit from a deeper exploration of how the findings can be applied in practical settings. The authors should elaborate on the implications for real-world engineering projects, particularly how these findings can be utilized to enhance rock instability predictions and safety measures in construction and mining industries.

7. Finally, while the study provides substantial insights into the behavior of different rocks under stress, the conclusion should more explicitly suggest future research directions. These could include investigations into the effects of variable environmental conditions on rock behavior, the application of different AE monitoring technologies, or the integration of the findings into predictive models for geological assessments.

Comments on abstract

- The abstract should specify the conditions under which the uniaxial compression and acoustic emission tests were conducted. Were there any specific environmental controls or variations in test setups across different rock types? Clarifying this could enhance the reproducibility and applicability of the research.

- The terms RA and AF are used without definitions. It's crucial for the authors to briefly define these parameters or provide references to their significance in the context of acoustic emission studies. This clarification will aid readers unfamiliar with these specific terms.

- While introducing lgN/b as a novel parameter, the authors should provide a justification or previous research basis for its inclusion. What makes lgN/b a valuable addition to the study, and how does it compare to or improve upon existing parameters?

- The claim that granite and siltstone have more complex failure mechanisms needs to be supported by data or references. The authors should elaborate on what makes the failure mechanisms of these rocks more complex, possibly through microstructural analysis or comparative studies.

- The relationship between the increase in shear cracks, the distribution of RA/AF values, and rock instability could be strengthened by linking these observations to existing theoretical frameworks or models in geotechnical engineering.

- The authors mention the utility of their findings for microseismic monitoring and warning strategies but should provide specific examples or case studies where their findings have been or can be applied. This addition would greatly enhance the real-world relevance of the research.

- Consider suggesting the inclusion of more specific keywords that reflect the novel aspects of the research, such as "microseismic precursors" or "rock damage characterization," to improve the searchability and impact of the paper.

Comments on Introduction

- While the introduction effectively links economic growth to increased infrastructure demands and the resulting geotechnical challenges, it would be beneficial to provide a more global perspective. Given that rock mechanics issues are not unique to any one country, broadening the discussion to include international examples or studies could enhance the relevance and applicability of the research.

- The introduction heavily cites recent studies from 2023 and 2024, which suggests a highly current research base. However, it's important to ensure that all cited works are accurately referenced and available for review, especially since some of the future-dated references may not yet be published. Clarifying these references or providing a note on forthcoming publications would improve credibility.

- The section mentions the use of uniaxial and triaxial compression tests along with acoustic emission technology but does not delve into why these specific methods were chosen over others. A brief discussion on the selection of these methods based on their proven efficacy in previous studies or their particular suitability for the rock types chosen would provide deeper insight into the research design.

- The authors discuss extracting precursor information for early warning systems but could further articulate how these findings translate into practical engineering applications. Specifically, discussing how early warning systems can be implemented at various construction sites and the potential impact on safety and cost-efficiency would strengthen the section.

- Ensure consistency in the terminology used to describe rock types and the processes studied. The terms "surrounding rock," "different type rocks," and others are used interchangeably. Standardizing these terms and providing clear definitions would enhance the readability and scientific rigor of the introduction.

- The introduction could benefit from a more explicit connection to foundational theories in rock mechanics. Linking current findings to established theories or models could provide a stronger theoretical framework for the study, reinforcing its scientific underpinnings.

- While the research objectives are mentioned, they could be stated more succinctly and prominently at the end of the introduction to clearly delineate the study's goals. This would help in setting the stage for the reader to understand the subsequent sections of the paper.

- To improve the Introduction, I suggest authors use following references

- https://doi.org/10.1186/s40703-017-0064-9

- https://doi.org/10.11113/jt.v76.4127

- https://doi.org/10.1016/j.measurement.2015.02.033

- DOI: 10.5772/intechopen.113218

- DOI: 10.32604/sdhm.2023.044573

- DOI: 10.5772/intechopen.112422

- https://doi.org/10.55121/nefm.v2i2.107

- https://doi.org/10.14382/epitoanyag-jsbcm.2021.13

- https://doi.org/10.1007/978-3-031-71097-1_9

- https://dx.doi.org/10.54203/jceu.2024.39

- https://dx.doi.org/10.54203/jceu.2023.10

- doi: 10.54517/ssd.v1i3.2314

- A Review of Clayey Soils, Asian Journal of Applied Sciences (ISSN: 2321 – 0893), Volume 04 – Issue 06, December 2016

- A Review of Physical and Chemical Clayey, Journal of Civil Engineering and Urbanism, Volume 6, Issue 4: 64-71; July 25, 2016

Comments on Test Design

- The description of the YAS-600 testing machine is quite detailed, which is excellent for replicability. However, it would enhance the paper to explain why this specific model was chosen over others, particularly what features make it ideal for the rocks being tested. Similarly, for the AE monitoring system, while the technical specifications are mentioned, a brief justification for the choice of trigger threshold, sampling rate, and filtering range based on previous literature or experimental requirements would strengthen the methodology.

- Reference to Fig.1 is made, which presumably illustrates the testing setup. It would be beneficial to encourage the authors to ensure that this figure is clear and detailed, possibly including labels or annotations that explain the setup thoroughly to aid in understanding without referring back to the text.

- The authors have done well in detailing the standardization of rock samples, including dimensions and machining accuracy. Suggesting that they emphasize the importance of these standards in the context of the tests' goals could highlight the rigor of the experimental design.

- The paper notes that three samples were tested for each rock type but only a typical sample was selected for analysis. The authors should clarify the criteria for selecting this "typical" sample and discuss how representative it is of the broader dataset. Additionally, they might consider addressing the implications of this selection for the generalizability of the results.

- While the non-parallelism of the sample end faces is controlled, it could be useful to prompt a discussion or acknowledgment of any other potential biases or limitations in the sample preparation or testing procedure that could affect the results. This transparency would lend credibility to the study.

- For Fig.2 which shows the rock samples, suggest that the authors provide a high-resolution image and consider including a scale or reference object for size comparison. This visual detail could help readers better understand the physical context of the samples discussed.

Comment on Analysis of AE parameters for different type rocks

- The use of RA/AF ratios and the introduction of lgN/b values are central to this analysis. It would be beneficial for the authors to provide a more detailed explanation or citation for the choice of these specific parameters and their thresholds. Specifically, the decision to use a k value of 0.05 as the cutoff for differentiating crack types should be supported by previous studies or empirical data.

- The references to various figures (Fig. 3 through Fig. 12) suggest that visual data plays a significant role in the analysis. It would be helpful for the authors to ensure these figures are not only clear and well-labeled but also accessible to readers in terms of color choices and scale. Encouraging the inclusion of a legend or more detailed captions could improve understanding.

- The detailed breakdown of crack proportion and RA/AF value distribution for each rock type is comprehensive. However, a comparative discussion section that directly contrasts these findings across the different rock types could enhance the reader's understanding of how these materials differ fundamentally in terms of mechanical behavior and AE response.

- The use of specific time periods and thresholds for warning points is intriguing. Encouraging the authors to include statistical validation of these warning points and discussing any models used to predict instability could lend more scientific rigor to the study. This might include sensitivity analysis or the testing of the model against known benchmarks.

- While the study is rich in technical detail, a clearer connection to practical applications would enhance its relevance. Suggest that the authors discuss how these findings can be implemented in real-world monitoring systems, perhaps by citing case studies or existing infrastructure projects where similar AE monitoring techniques have been successfully applied.

- The analysis seems robust, but any scientific study has limitations. It would strengthen the paper if the authors could discuss the limitations of their approach—such as the potential for different results under variable environmental conditions or in larger-scale studies—and suggest areas for future research.

- Ensure that the use of technical terms and the presentation of data are consistent throughout the document. Encouraging the authors to review the uniformity of terms like "time periods," "stages," and "phases" and their definitions could prevent confusion.

Comments on Discussion

- The conclusion effectively summarizes how different rock types respond under uniform testing conditions, highlighting the distinct mechanical behaviors and AE characteristics. Encourage the authors to briefly restate the significance of these findings in the broader context of geotechnical research and their implications for rock mechanics and engineering.

- The explanation of how different rocks behave under stress is clear, but it could be enhanced by directly linking these observations to practical implications for engineering design, such as how these findings might influence the selection of materials in construction projects in seismically active areas.

- The authors mention that the experiments were conducted under a constant loading rate, which is an important detail. Suggest that they discuss the potential effects of varying loading rates on the AE characteristics and rock behavior, which could provide insights into the robustness of their conclusions across different stress scenarios.

- The paper discusses the application of hierarchical warnings based on AE parameters. The authors could expand on how these warning systems can be integrated into existing geological and construction practices. Suggest including a discussion on the interoperability of these systems with other monitoring technologies or methods.

- While managing timing and establishing thresholds for warnings are highlighted as challenges, encourage the authors to elaborate on specific technological or methodological advancements that might help overcome these challenges. For instance, could advanced computational models or machine learning techniques improve the precision or reliability of these warnings?

- Ensure that technical terms, especially those describing complex processes like the instability mechanisms and the function of AE parameters, are consistently defined and used throughout the paper. This consistency will help maintain clarity and enhance the paper's professional tone.

- The conclusion is an appropriate place to acknowledge any limitations of the study. Encourage the authors to specify any constraints they encountered in the experimental design or analysis and suggest areas for future research, such as testing under variable environmental conditions or scaling up the experiments.

**Reviewer #6:**  Dear Authors,

I hope you're doing well. I’ve reviewed your paper titled "Instability Mechanisms and Precursor Information in Different Rock Types Based on Acoustic Emission," and I would like to share some feedback that might help improve the paper.

First, your study provides valuable information about rock instability and acoustic emission, which is important for many industries. Your methodology is solid, and the results are useful. However, I have a few suggestions for improvement:

Comments to Author

Abstract

1. What’s Good:

The abstract is a good start, but it could be simplified.

You mention the key findings, but some technical terms (like "RA/AF values" and "lgN/b values") might be confusing for a wider audience. A quick explanation of these terms would help.

2. Suggestions:

Try not to start sentences with too much technical jargon. For example, instead of starting with "Detailed analysis of AE parameters...," give some context first.

o Break up longer sentences so the main points come across more clearly.

3. Comment: Try to simplify the abstract and introduce the technical terms in a way that's easy for a broader audience to understand. Make sure the main findings and methods are clear.

Introduction

4. What Could Be Better?

A. The flow is a little choppy because of all the citations. It feels like the references are clustered together and might overwhelm the reader.

b. There’s no clear mention of the research gap, which would help highlight what makes this study unique.

5. Suggestions:

a. Consider combining citations to make the introduction smoother (e.g., instead of listing multiple references, say "previous studies show...").

b. Clearly state the research gap. What’s missing in the current research, and how does your study fill that gap?

c. Add a simple paragraph summarizing what your study contributes to the field in a straightforward way.

6. Comment: Clarify the research gap and simplify the language to make it accessible for a wider range of readers. It’ll help set up the rest of the paper better.

Methodology

7. Suggestions:

a. Make sure to include enough detail so others can replicate the experiment—especially things like AE monitoring settings and how you processed the data.

b. If possible, include a diagram of your experimental setup to make it easier to understand.

c. It might be helpful to mention any limitations of the methodology and how you dealt with them.

Discussion

1. Lack of Comparison: The discussion would be stronger if you compared your findings with those of other studies. Are your results similar to or different from what others have found?

i. Suggestion: Include some references to earlier studies so readers can see how your results fit into the broader research.

2. No Citations: Right now, the discussion doesn’t reference other studies to support your conclusions.

i. Suggestion: Adding citations will help back up your findings and show how your work fits with previous research.

3. Limited Practical Implications: You mention interesting findings, but it would be great to see how these could be applied in real-world situations like construction or mining.

i. Suggestion: Discuss how your findings could help improve safety or stability in those industries and what challenges might come up when applying them in the field.

4. No Discussion of Limitations: It’s important to mention any limitations of your experiments, like how experimental conditions (e.g., loading rate, sample size) could have influenced the results.

i. Suggestion: Acknowledge those limitations and explain how they might affect your findings.

5. No Suggestions for Future Research: It’s always good to suggest what could be explored next.

i. Suggestion: Highlight areas for future studies—maybe further research on other types of rocks or developing better warning systems for rock instability.

6. Comment: Strengthen the discussion by comparing your findings to others, addressing practical uses of your results, acknowledging limitations, and suggesting next steps for research.

Conclusion

You wrap up the main points well and emphasize the potential for multi-parameter warning systems.

1. What Could Be Better?

o Repetitiveness: The conclusion repeats a lot of what’s already in the discussion. It could be more focused.

o Suggestion: Try to make the conclusion shorter and focus on the most important points.

2. Missed Bigger Picture: The conclusion doesn’t really tie your findings to the broader field of rock mechanics or engineering. It would be nice to see how your work could influence those areas.

Suggestion: Explain how your findings can make a difference in these fields or improve current methods.

3. No Actionable Steps: While you talk about warning systems, there aren’t clear steps for implementing them.

Suggestion: Add specific recommendations on how to use your findings in the field, like creating protocols for monitoring rock instability.

General Comments

1. Language and Grammar:

There are a few grammatical errors and some language that’s too technical, which might make the paper harder to read for those not familiar with the field.

Try to keep the tense consistent throughout the paper, especially when describing your findings.

Example Fix:

Original: "The study results showed that an increase in the proportion of shear cracks...".

Revised: "The results show that an increase in shear cracks..."

6. PLOS authors have the option to publish the peer review history of their article (what does this mean? ). If published, this will include your full peer review and any attached files.

**Do you want your identity to be public for this peer review?** For information about this choice, including consent withdrawal, please see our Privacy Policy .

Reviewer #1: No

Reviewer #2: No

Reviewer #3: No

Reviewer #4: No

Reviewer #5: **Yes: ** Ali Akbar Firoozi

Reviewer #6: No

---

## [Author Response · Author response to Decision Letter 1]

12 Mar 2025

Revision Note

Dear editor and reviewers:

First of all, the authors would like to thank you for the constructive comments and valuable recommendations on the manuscript entitled “The instability mechanisms and precursor information of different type rocks based on acoustic emission”. The authors have reviewed comments and recommendations carefully, and we have made corresponding corrections. The revisions are marked red in the manuscript and the point-by-point responds to the editor’s and the reviewers’ comments are as follows:

Comments from the six reviewers:

Reviewer #1:

This study investigated the instability mechanisms and precursor information of five rock types—marble, granite, siltstone, sandstone, and limestone—based on uniaxial compression tests combined with acoustic emission (AE) monitoring. However, enhancing the clarity of the methodology, broadening the discussion's practical context, and providing further justification for certain methodological choices would strengthen the paper.

Q1-Condense the abstract by focusing on key findings and their significance. The current version is dense and includes excessive technical details, which could be streamlined to enhance readability.

Response: Thank you very much for your comments. We have revised them.

Q2-Highlight the novelty of your study more explicitly. What sets your research apart from previous studies on rock instability and acoustic emission monitoring?

Response: Thank you very much for your comments. The instability laws and monitoring parameter thresholds of surrounding rock vary at different construction sites. In the early stage of construction, in order to better serve on-site construction, it is necessary to understand the instability process and mechanism of different type surrounding rocks in advance, and then formulate monitoring and warning plans. The paper aims to explore the same laws and different characteristics of instability processes and precursor information for different type rocks. Five different type rocks from various sources were selected for research and analysis. Furthermore, the microseismic parameter lgN/b was introduced for the first time. Analyzing the lgN/b values dynamic change process revealed the instability laws and rock instability precursor information was obtained for different type rocks. Additionally, the study found that the change laws of lgN/b values can reveal the extent of development of intrinsic weak structures inside the rocks. Finally, through a multi-parameter and multi-method comprehensive analysis, we can more scientifically identify the instability precursor information of different type rocks in terms of time, and the hierarchical warning was achieved.

Q3-Provide more detail on the rationale behind using RA/AF and lgN/b parameters for instability prediction. Briefly explain why these metrics are superior or complementary to other methods.

Response: Thank you very much for your comments.

(1) Numerous studies (Wang et al. 2021; Luo et al. 2024) have shown that the RA and AF ratio can effectively characterize the types of rock damage and reveal the mechanisms of rock instability. A low AF value and a high RA value indicate the generation and development of shear cracks, whereas the opposite conditions suggest the generation and development of tensile cracks. This study employed the RA/AF ratio (k) to differentiate between tensile and shear cracks. When the value of k exceeds 0.05, it signifies the formation of shear cracks inside the rock; otherwise, it indicates the generation of tensile cracks (Luo et al. 2024).

References

1. Wang Y, Zhang B, Gao SH, Li CH. Investigation on the effect of freeze-thaw on fracture mode classification in marble subjected to multi-level cyclic loads. Theor Appl Fract Mech. 2021; 111:102847. https://doi.org/10.1016/j.tafmec.2020.102847

2. Luo DN, Hu ZK, Shi Y, Qing LB, Su GS. Experimental study on biaxial mechanical properties and acoustic characteristics of different damage granite after high temperature-water cooling. Chin J Rock Mech Eng. 2024; 43(07):1680-1695.

(2) The b value represents the proportional relationship between the number of large magnitude events and the number of small magnitude events. Therefore, the change in the b value can be used to reflect the change in the stress field of the surrounding rock mass. The larger the number of events N, the more microcracks will appear in the surrounding rock mass, and the greater the risk of surrounding rock mass instability. The lgN/b value can represent the number of large magnitude events. By analyzing the lgN/b values of the five type rocks mentioned above, it can be concluded that the lgN/b values can effectively reflect the internal AE activity characteristics of rocks from the initial loading to instability, and can well characterize the process of micro-fracture generate, development, and expansion inside rocks. The warning effect was good.

Q4-While the introduction explains the significance of the study, it could benefit from more in-depth references to recent research in the past five years. Highlight gaps in the literature this study addresses, such as: compgeo. 2024.106095; tafmec.2024.104691; fuel.2023.129584

Response: Thank you very much for your comments. We have added these recent papers as references.

References

1. Li ZX, Fujii Y, Alam AKMB, Li ZH, Du F, Wei WJ. Implementing a simple 2D constitutive model for rocks into finite element method. Comput Geotech. 2024; 167:106095. https://doi.org/10.1016/j.compgeo.2024.106095

2. Wang K, Chang CG. Study on the characteristics of CO2 fracturing rock damage based on fractal theory. Theor Appl Fract Mech. 2024; 134: 104691. https://doi.org/10.1016/j.tafmec.2024.104691

3. Wang K, Pan HY, Fujii Y. Study on energy distribution and attenuation of CO2 fracturing vibration from coal-like material in a new test platform. Fuel. 2024; 356:129584. https://doi.org/10.1016/j.fuel.2023.129584

Q5-Clarify the selection criteria for the five rock types. Was it based on geographic diversity, mechanical properties, or application relevance?

Response: Thank you very much for your comments. It was based on geographic diversity and application relevance.

Q6-The results are well-documented but could benefit from clearer visual aids. Simplify or reorganize some graphs for better comprehension.

Response: Thank you very much for your comments. We have revised them for better comprehension.

Q7-Compare your findings with similar studies in more detail, emphasizing agreement or divergence. This contextualization will strengthen the discussion.

Response: Thank you very much for your comments. We have revised them.

Q8-Reiterate the practical significance of your findings, such as how they might influence on-site monitoring or the design of safety protocols.

Response: Thank you very much for your comments. The instability laws and monitoring parameter thresholds of surrounding rock vary at different construction sites. In the early stage of construction, in order to better serve on-site construction, it is necessary to understand the instability process and mechanism of different type surrounding rocks in advance, and then formulate monitoring and warning plans. The paper aims to explore the same laws and different characteristics of instability processes and precursor information for different type rocks. Through a multi-parameter and multi-method comprehensive analysis, we can more scientifically identify the instability precursor information of different type rocks in terms of time, and the hierarchical warning was achieved. We can take timely measures to save costs and improve construction efficiency. The research results can provide a reference for microseismic monitoring at various sites and establish certain scientific basis for effective on-site warning strategies.

Q9-Discuss the limitations of your study and potential future work, such as testing under field conditions or extending the methodology to other rock types.

Response: Thank you very much for your comments. The on-site environment is very complex, and sensors can receive a lot of noise and the signal also contains some noise; The laboratory environment is relatively simple. The rock damage also includes unloading damage on site. Blasting shock and structural planes can also affect the failure mechanism of rocks. All of the above are the direction and focus of our future research.

Reviewer #2:

This manuscript studies the instability mechanism and precursor information of different types of rocks based on acoustic emission. This topic is very important, and the author has meaningful insights on it. The structure of the manuscript is reasonable, and the content is also rich, but the research depth is slightly insufficient. Minor revision is recommended.

Q1-There are some grammatical errors in this manuscript, and an improvement of the English editing is needed.

Response: Thank you very much for your suggestions. We have checked the technical writing and polished the language with the help of a professor with good English writing.

Q2-What is the sensor type selected for the AE system?

Response: Thank you very much for your suggestions. The selected AE sensor type is R6α, which has high sensitivity.

Q3-Please explain the meaning of RA and AF values when their first appearance in the manuscript.

Response: Thank you very much for your suggestions. We have explained the meaning of RA and AF values.

RA=(rise time)/amplitude

AF=(AE counts)/duration

Q4-Please explain the meaning of “N” and “n” in the manuscript. Do the two letters have the same meaning?

Response: Thank you very much for your suggestions. n is the number of AE hits whose amplitude is no less than AdB/20 in every time window; N is the total number of AE hits in every time window. The two letters have the different meaning?

Q5-The author's understanding of the rock fracture mechanism is somewhat lacking. It is recommended to refer to the following literature for further analysis

Analytical solution of the stress field and plastic zone at the tip of a closed crack. FRONTIERS IN EARTH SCIENCE. Doi: 10.3389/feart.2024.1370672

Response: Thank you very much for your comments. We have added the recent paper as a reference.

References

1. Wu GZ, Wang WS, Peng SC. Analytical solution of the stress field and plastic zone at the tip of a closed crack. Front Earth Sci. 2024; 12:1370672. https://doi.org/10.3389/feart.2024.1370672

Reviewer #3:

Authors have meaningful insights into the topic. This is a research work with practical engineering significance. It is my opinion that this manuscript can be accepted for publication after considering the following issues.

Q1-The abstract should accurately reflect the research purpose, research methods, research results, and conclusions of the paper. It is recommended to rewrite it.

Response: Thank you very much for your comments. We have revised them.

Q2-Although the introduction provides a literature review, it is not easy to truly grasp the work's novelty for the experimental study on rock mechanics. The aims and novelties should be explicitly mentioned in the introduction. It is suggested that the author supplement the research results on RA/AF and b values of rocks.

Response: Thank you very much for your comments. We have supplemented the research results on RA/AF and b values of rocks.

References

1. Liu XL, Han MS, He W, Li XB, Chen DL. A New b Value Estimation Method in Rock Acoustic Emission Testing. J Geophys Res Solid Earth. 2020; 125: e2020JB019658. https://doi.org/10.1029/2020JB019658

2. Liu XL, Liu Z, Li XB, Gong FQ, Du K. Experimental study on the effect of strain rate on rock acoustic emission characteristics. Int J Rock Mech Min Sci. 2020; 133: 104420. https://doi.org/10.1016/j.ijrmms.2020.104420

3. Sun B, Yan Y, Wang SY, Qi CM, Yang HW, Zeng S. Self-Organized Criticality and b-Value Characteristics of Acoustic Emission of Rocks Under Different Stress Paths. Rock Mech Rock Eng. 2025; 58, 851–866. https://doi.org/10.1007/s00603-024-04191-z

Q3-The key words selected in the paper are not accurate enough and representative, for example, rock mechanics, uniaxial compression, acoustic emission characteristics, early warning.

Response: Thank you very much for your comments. We have revised them.

Keywords: Rock; Acoustic emission; RA/AF values; lgN/b values; Early warning

Q4-In line 96,“The equipment is currently recognized as one of the most advanced and feature-rich uniaxial testing machines for rocks available in China”, I do not recommend evaluating experimental equipment.

Response: Thank you very much for your comments. We have revised them.

Q5-The full text suggests the author to refine the language expression, such as the first and third paragraphs of the introduction, equipment description, etc.

Response: Thank you very much for your comments. We have revised them.

Q6-The parameter Settings of the acoustic emission acquisition system, such as sensor resonant frequency, HLT, HDT, PDT, etc., should also be provided.

Response: Thank you very much for your comments. We have provided them.

Table 1 Parameters of acoustic emission sensors

Threshold/dB sampling rate/ MSPS PDT/us HDT/us HLT/us

40 5 50 200 300

Q7-In Fig.4, Please explain that there is no second warning point for granite.

Response: Thank you very much for your comments. To make up for our omission, we have added the second warning point for granite.

Q8-Why did Fig.8, Fig.9, and Fig12 fail to get the second warning point? Does it indicate that the sliding window selected by the author is too large, resulting in inaccurate calculation results of b value?

Response: Thank you very much for your comments. To make up for our omission, we have added the second warning point in Fig.8 and Fig12. In Fig.9, the point with a significant increase is also the peak point, so we only set one warning point.

Q9-The first warning points of marble and siltstone are different under the two methods, what is the reason?

Response: Thank you very much for your comments. The inconsistent warning effects of the two AE parameters are due to the complexity of the AE signal itself, and different parameters may reflect different aspects of material damage. For AE technology, different AE parameters may have different sensitivity and specificity, resulting in inconsistent warning effects for the same damage process.

Q10-It is suggested that when analyzing the variation characteristics of RA/AF value and lg N/b, the author finally summarizes, analyzes and compares the difference of the precursor law of failure of five different lithologies.

Response: Thank you very much for your comments. We finally summarized, analyzed and compared the difference of the precursor law of failure of five different lithologies.

Q11-There are some grammatical errors in this manuscript, and an improvement is needed.

Response: Thank you very much for your suggestions. We have checked the technical writing and polished the language with the help of a professor with good English writing.

Q12-“lgN/b values can effectively reflect the internal AE activity characteristics of rocks”. What is the physical meaning of the parameter?

Response: Thank you very much for your suggestions. The b value represents the proportional relationship between the number of large magnitude events and the number of small magnitude events. Therefore, the change in the b value can be used to reflect the change in the stress field of the surrounding rock mass. The larger the number of events N, the more microcracks will appear in the surrounding rock mass, and the greater the risk of surrounding rock mass instability. The lgN/b value can represent the number of large magnitude events.

Q13-Section “Analysis of AE parameters for different type rocks”. The meaning of RA and AF values need to be explained.

Response: Thank you very much for your suggestions. We have explained the meaning of RA and AF values in the revised manuscript.

RA=(rise time)/amplitude

AF=(AE counts)/duration

Q14-How to achieve synchronization for time control of AE monitoring system and time control of the loading system?

Response: Thank you very much for your suggestions. Two people c

---

## [Decision Letter · Decision Letter 1]

18 Mar 2025

The instability mechanisms and precursor information of different type rocks based on acoustic emission

PONE-D-25-00922R1

Dear Dr. Xue,

We’re pleased to inform you that your manuscript has been judged scientifically suitable for publication and will be formally accepted for publication once it meets all outstanding technical requirements.

Kind regards,

Kang Wang, Ph.D.

Academic Editor

PLOS ONE

Additional Editor Comments (optional):

Reviewers' comments:

Reviewer's Responses to Questions

**Comments to the Author**

1. If the authors have adequately addressed your comments raised in a previous round of review and you feel that this manuscript is now acceptable for publication, you may indicate that here to bypass the “Comments to the Author” section, enter your conflict of interest statement in the “Confidential to Editor” section, and submit your "Accept" recommendation.

Reviewer #2: All comments have been addressed

Reviewer #3: (No Response)

Reviewer #5: All comments have been addressed

2. Is the manuscript technically sound, and do the data support the conclusions?

Reviewer #2: Yes

Reviewer #3: (No Response)

Reviewer #5: Yes

3. Has the statistical analysis been performed appropriately and rigorously? 

Reviewer #2: Yes

Reviewer #3: (No Response)

Reviewer #5: Yes

4. Have the authors made all data underlying the findings in their manuscript fully available?

Reviewer #2: Yes

Reviewer #3: (No Response)

Reviewer #5: Yes

5. Is the manuscript presented in an intelligible fashion and written in standard English?

Reviewer #2: Yes

Reviewer #3: (No Response)

Reviewer #5: Yes

6. Review Comments to the Author

Reviewer #2: I think the manuscript has been carefully revised and is ready for acceptance. I have no more comments.

Reviewer #3: Here are two suggestions:

(1) Carefully check the format and non-standard writing of the manuscript to meet the publication requirements of the journal.

(2) The conclusion needs to be clearer and some unnecessary descriptions should be removed.

Reviewer #5: The manuscript is acceptable, and the author has addressed the comments provided. No concerns regarding dual publication, research ethics, or publication ethics were noted.

7. PLOS authors have the option to publish the peer review history of their article (what does this mean? ). If published, this will include your full peer review and any attached files.

**Do you want your identity to be public for this peer review?** For information about this choice, including consent withdrawal, please see our Privacy Policy .

Reviewer #2: No

Reviewer #3: No

Reviewer #5: **Yes: ** Ali Akbar Firoozi, PhD

---

## [Editor Report · Acceptance letter]

PONE-D-25-00922R1

PLOS ONE

Dear Dr. Xue,

I'm pleased to inform you that your manuscript has been deemed suitable for publication in PLOS ONE. Congratulations! Your manuscript is now being handed over to our production team.

Kind regards,

on behalf of

Dr. Kang Wang

Academic Editor

PLOS ONE